# A UAV-Based Air Quality Evaluation Method for Determining Fugitive Emissions from a Quarry during the Railroad Life Cycle

**DOI:** 10.3390/s21093206

**Published:** 2021-05-05

**Authors:** Min-kyeong KIM, Yelim Jang, Jaeseok Heo, Duckshin Park

**Affiliations:** 1Railroad Test & Certification Division, Korea Railroad Research Institute, Uiwang 16105, Korea; mkkim15@krri.re.kr; 2Transportation Environmental Research Team, Korea Railroad Research Institute, Uiwang 16105, Korea; yelimm412@krri.re.kr (Y.J.); jsheo1005@krri.re.kr (J.H.); 3Department of Transportation System Engineering, University of Science & Technology (UST), Daejeon 34113, Korea

**Keywords:** unmanned aerial vehicle (UAV), air quality evaluation, railroad life cycle, quarry, fugitive dust

## Abstract

Gravel is used in railway infrastructure to reduce environmental impacts and noise, but gravel on tracks must be replaced continuously because it deforms due to wear and weathering. It is therefore necessary to review the entire railroad life cycle. In this study, an unmanned aerial vehicle (UAV) was used to measure resuspended dust over a wide area. The dust was generated from transport movements in relation to the operation of a quarry, which represents the first stage of the railway life cycle. The dust was measured at Gangwon-do quarry using a Sniffer4D module, which can provide measurements at 1 s intervals through a light scattering method and has high reliability (R^2^ = 0.95 for PM_2.5_, R^2^ = 0.88 for PM_10_). The hourly generation of fugitive dust was calculated as 2937.5 g/h for PM_2.5_ and 4293.2 g/h for PM_10_. The social cost of dust generation was calculated as KRW 36.59 billion. The amount of dust generated per hour at the quarry was ~12 times greater than that generated by the operation of a regulator as a maintenance vehicle, with the largest amount of fugitive dust generated by the washing-type vehicle. This is the first study to measure the amount of fugitive dust generated in real time at 1 s intervals by monitoring the first stage of the railroad life cycle over a wide area using a Sniffer4D module attached to a UAV. This method can be replicated for use in various studies.

## 1. Introduction

Human health can be compromised by exposure to particulate matter (PM) for long periods of time, with immunity rapidly lowered, leading to various conditions such as cardiovascular, skin, and eye diseases, as well as respiratory diseases such as colds, asthma, and bronchitis [1,2,3,4,5]. In particular, black carbon is emitted as fine dust from diesel engines and has been categorized as a first-class carcinogen by the World Health Organization, while governments allow the emission of certain levels of sulfur oxides and nitrogen oxides that can create secondary PM [6,7]. Since 2015, the PM air quality standard in Korea has been strengthened by 20–25%, and a standard for fine dust emission from gasoline engines has also been recently established. When reviewing the air environment standards, the average value of PM_10_ for 24 h is less than 100 µg/m^3^. In addition, PM_2.5_ is set at a 24 h average value of 35 µg/m^3^ or less, and an annual average value of 15 µg/m^3^ or less in Korea [6]. In addition, emission limits for air pollutants discharged from diesel locomotives were established in the Air Environment Conservation Act, and diesel locomotives manufactured and imported into Korea from 2020 onwards must comply with the newly established emission permit standards. The management of fine dust emission sources throughout the railway industry has also been strengthened [6].

Interest in PM pollution problems in Korea has steadily increased, and a move toward sustainable railroad construction and the improvement of railroad facilities has been proposed [7,8]. The gravel tracks constructed to date in Korea have followed a traditional structure and have been widely used due to their relatively low noise production and low initial construction costs. However, gravel is scattered on the road during construction and repair, which generates PM [7,9,10]. Fugitive dust emissions are therefore generated during gravel track maintenance. Fugitive dust can also be generated from aggregated storage piles [11], paved and unpaved roads [12], heavy construction [13], dust storms [14,15], and agricultural operations [16,17,18]. This causes various environmental problems, such as the occurrence of waste gravel and damage to the natural environment due to the operation of quarries.

Concrete roads have been proposed as an alternative to gravel tracks because they generate less PM during construction and repair work, but there have been no studies to quantify the air pollutant emissions from the two types of road. Previous studies have investigated the amount of PM generated on gravel and concrete roads in specific areas using gravimetric and light scattering methods. However, no research has been conducted that comprehensively considers the railroad life cycle, such as the operation of quarries, maintenance vehicles, and train operation.

In this study, measurements were limited to the amount of fine dust generated in the operation of a quarry, which represented the first stage of the railway life cycle. Because quarries cover an extensive area and are difficult to access by humans, an unmanned aerial vehicle (UAV)-based measurement system was adopted. A particle aerosol spectrometer (1.108, 1.109; Grimm Aerosol Technik, Ainring, Germany), which is a light scatter measuring instrument [19,20,21], can be used to measure the amount of fugitive dust generated in such areas; however, in this study, real-time measurements were made at 1 s intervals using a Sniffer4D module attached to a UAV. This represented a novel measurement system for the collection of air pollution data. As a recent study using Sniffer4D, there is a case in which a steel manufacturer (POSCO Plant) located on the coast has conducted a study on reduction measures by quantifying the induction of particulate matter in iron ore and coal caused by strong winds. In addition, a study using Sniffer4D was carried out for the rapid monitoring of air pollutants in the management area from the local governments to the particulate matter management area in Korea. However, this is the first time that a Sniffer4D sensor has been attached to a UAV to measure fine dust generated in a quarry.

In this study, the amount of fugitive dust generated from the operation of a quarry was determined by considering the operation of a maintenance vehicle on gravel tracks. A comparative analysis of the emissions produced was performed. The amount of fine dust generated during the operation of a quarry, the first stage of the railroad life cycle, was then derived, and the emission rate over the entire railroad life cycle was estimated. This study developed a new UAV-based method to measure the PM concentration over a wide area in real time, which can also be used in other fields.

## 2. Materials and Methods

### 2.1. Study Site

To obtain basic data for the estimation of the amount of fine PM generated by the operation of quarries using gravel as a raw material on gravel tracks, two quarries were selected, one located in Wonju-si and the other in Gangwon-do (Figure 1). In both locations, it was possible to make measurements with the UAV, and gravel tracks were targeted. A preliminary survey was conducted using Google Earth to select areas near the quarries that could be used to determine background concentrations. Unpolluted areas in the vicinity of the quarries, including rice paddies and fields, were selected at distances of ~1–2 km away. To select the study regions, the entire area was reviewed using Google Earth to provide basic data. At the target sites, the weather forecast was monitored and the nearby areas were measured on 10 September 2020. In this study, meteorological parameters were considered. The wind direction at around 2:00 pm, measured at X quarry mine and nearby areas, was 305.3 deg. In addition, the wind direction at around 4:00 pm, measured at Y quarry mine and nearby areas, was 290 deg.

### 2.2. Experimental Methods 

A Matrice-300 RTK aircraft (DJI, Shenzhen, China) was used to acquire time-series aerial images of the target site (Figure 2). This aircraft is a rotary-wing UAV capable of taking off and landing vertically and is suitable for areas with a narrow flight location. It has the advantage of acquiring precise images at a lower altitude than a fixed wing aircraft. Using a Sniffer4D 1043578b (module id 100) equipped with a light scattering sensor, the fugitive dust concentration generated from the quarry was measured at 1 s intervals.

A large area was measured to estimate the amount of fugitive dust generated during operation of the quarry. First, the area of the quarry was estimated. From the spatial characteristics of the quarry, a large number of measurement locations were required and it was difficult for individuals to reach them. The UAV was therefore used for all measurements.

### 2.3. Reliability of the Sniffer4D Sensor

The reliability of the Sniffer4D sensor, which measures the amount of PM generated by a light scattering method, was confirmed. These tests were conducted by the Department of Environmental Engineering at Jinan University in China (Guangzhou), and were designed to compare the Thermo Scientific Super Station and Sniffer4D sensors for around 180 days to evaluate the long-term data correlation between the two monitoring methods for particulate matter concentrations. Correlation coefficients of R^2^ = 0.95 for PM with a diameter of 2.5 μm or less (PM_2.5_) and R^2^ = 0.88 for PM_10_ were obtained (Figure 3).

The method using the Sniffer4D module was therefore considered highly reliable and could be confidently used as light scattering method for measuring PM. The equipment was then used in this study.

## 3. Results and Discussion 

### 3.1. Air Quality Measurements Using the UAV-Based Method

#### 3.1.1. Measurement of the PM Concentration at X Quarry and Nearby Areas

During the operation of X quarry, the UAV approached the point of fugitive dust emissions and the PM concentration was measured. The total measurement area of X quarry was 42,799 m^2^, and the average grid size of the area that could be detected by the dust measurement sensor was 39.8 m (width) × 39.8 m (length), resulting in a measurement area of 1585.1 m^2^. The measurements were conducted over a period of 13 min and 39 s (from 13:56:48 to 14:10:47 on 10 September), and a total of 839 datapoints were collected (Figure 4).

The locations used to measure the background concentration for X quarry were 1.7 km from the quarry and were selected because they were not affected by the quarry according to the preliminary investigation. The total measurement area was 90,336.8 m^2^ and the average grid size of the area that could be detected by the dust measurement sensor was 39.8 m (width) × 39.8 m (length), resulting in an area of 1584.9 m^2^. Measurements were performed over a period of 33 min 52 s (from 14:21:25 to 14:55:17 on 10 September), and a total of 2032 datapoints were collected (Figure 4).

The average PM_2.5_ concentration was 0.028 mg/m^3^, with the highest and lowest concentrations among the average values being 0.032 and 0.025 mg/m^3^. At a single point, the maximum concentration of 0.033 mg/m^3^ occurred at 14:03:35, and the minimum of 0.024 mg/m^3^ occurred at 14:06:22. The average PM_10_ concentration was 0.030 mg/m^3^, with the highest and lowest concentrations among the average values being 0.034 and 0.027 mg/m^3^. At a single point, the maximum concentration of 0.038 mg/m^3^ occurred at 14:03:27, and the minimum of 0.026 mg/m^3^ occurred at 14:06:22 (Figure 5).

When measuring the fugitive dust in the vicinity of X quarry, the average PM_2.5_ concentration was 0.027 mg/m^3^, with the highest and lowest concentrations among the average values being 0.032 and 0.025 mg/m^3^. At a single point, the maximum concentration of 0.033 mg/m^3^ occurred at 14:03:35, and the minimum of 0.022 mg/m^3^ occurred at 14:46:20. The average PM_10_ concentration was 0.029 mg/m^3^, with the highest and lowest concentrations among the average values being 0.034 and 0.026 mg/m^3^. At a single point, the maximum concentration of 0.038 mg/m^3^ occurred at 14:03:27, and the minimum of 0.024 mg/m^3^ occurred at 14:45:53 (Figure 6).

#### 3.1.2. Measurement of the PM Concentration at Y Quarry and Nearby Areas

During the operation of Y quarry, the UAV approached the point of fugitive dust emissions, and the PM concentration was measured. The total measurement area of Y quarry was 17,435 m^2^, and the average grid size of the area that could be detected by the sensor was 39.8 m (width) × 39.8 m (length), resulting in a measurement area of 1585.0 m^2^. The measurements were conducted over a period of 9 min and 19 s (from 16:15:34 to 16:24:53 on 10 September), and a total of 559 datapoints were collected (Figure 7).

The locations used to measure the background concentration for Y quarry were 1.1 km away and were selected because they were not affected by the quarry according to the preliminary investigation. The total measurement area was 55,480 m^2^, and the average grid size of the area that could be detected by the sensor was 39.8 m (width) × 39.8 m (length), resulting in a measurement area of 1585.0 m^2^. The measurements were conducted over a period of 9 min and 40 s (from 16:48:30 to 16:58:10 on September 10), and a total of 580 datapoints were collected (Figure 7).

The average PM_2.5_ concentration was 0.038 mg/m^3^, with the highest and lowest concentrations among the average values being 0.043 and 0.027 mg/m^3^. At a single point, the maximum concentration of 0.061 mg/m^3^ occurred at 16:24:22, and the minimum of 0.025 mg/m^3^ occurred at 16:17:35. The average PM_10_ concentration was 0.047 mg/m^3^, with the highest and lowest concentrations among the average values being 0.056 and 0.029 mg/m^3^. At a single point, the maximum concentration of 0.083 mg/m^3^ occurred at 16:23:35, and the minimum of 0.028 mg/m^3^ occurred at 16:17:32 (Figure 8).

The concentration of fugitive dust in the vicinity of Y quarry was measured and used as a background concentration. The average PM_2.5_ concentration was 0.026 mg/m^3^, with the highest and lowest concentrations among the average values being 0.030 and 0.024 mg/m^3^. At a single point, the maximum concentration of 0.031 mg/m^3^ occurred at 16:52:31, and the minimum of 0.021 mg/m^3^ occurred at 16:50:16. The average PM_10_ concentration was 0.028 mg/m^3^, with the highest and lowest concentrations among the average values being 0.033 and 0.025 mg/m^3^. At a single point, the maximum concentration of 0.033 mg/m^3^ occurred at 16:52:31, and the minimum of 0.024 mg/m^3^ occurred at 16:50:30 (Figure 9).

### 3.2. The Amount of Fugitive Dust Generated as Part of the Railroad Life Cycle

In this study of the railroad life cycle, a UAV-based method was used to measure the fugitive dust that was generated during the operation of a quarry. In X quarry, water was continuously sprayed to reduce the PM concentration, and it was therefore difficult to determine the PM concentration throughout the operation of the quarry.

The amount of PM generated by fugitive dust emissions during the operation of the quarry was calculated for Y quarry. The measurement area was 17,435 m^2^, which is the total area of Y quarry, and the average flow velocity was 3.6 m/s (i.e., the wind speed at Wonju-si station at 16:00) at the time the fugitive dust emissions were measured in Y quarry and at locations nearby Y quarry. The wind speed data used here were Meteorological Administration public data. According to Watson and Chow (2000) [22], the rate of dust discharge from erosive surfaces is affected by wind speed. Fugitive dust emission fluxes are typically calculated using a fugitive dust model in an iterative manner before being fitted to a power function that displays a wind velocity dependence [23]. Because the relaxation time of particles 10 µm or smaller in the general atmosphere is very short, fugitive dust is predominantly affected by the ambient air velocity [24]. The average flow velocity in Wonju-si was 3.6 m/s, and the maximum possible wind speed for the UAV model used in the study was 15 m/s. The suction port of the Sniffer4D module introduced air at a constant velocity, and therefore it was easy to measure the PM concentration.
(1)M=A×V×C×n

In Equation (1), M is the mass of PM generated per unit time (mg/s), A is the area of occurrence (m^2^), V is the flow rate (m/s) that generates fine dust, C is the average concentration of fine dust (mg/m^3^), and n is the number of fugitive dust generation areas [7].

The amount of dust generated was calculated by correcting the average concentration for each PM size class in Y quarry to the average concentration for each PM size class in the background area (i.e., control) for Y quarry. The hourly generation of fugitive dust was calculated to be 2937.5 g/h for PM_2.5_, and 4293.2 g/h for PM_10_ (Table 1).

Currently, 28 offices are using maintenance equipment, and looking at the annual operation rate for each equipment, a total of 26,494 h is used for the multiple tie tamper, 18,334 h for the regulator, and 4418 h for the ballast cleaver in Korea. When reviewing the entire railroad life cycle, the PM_10_ generation rate of a typical maintenance vehicle on gravel tracks was 190.5 g/h for the operation of a multiple tie tamper, 358.1 g/h for a regulator, and 191.0 g/h for a ballast cleaver [7]. Comparing this with Y quarry, it was apparent that the amount of dust generated per hour in Y quarry was around 12 times that generated by a regulator, with the greatest amount of fugitive dust being generated by the washing-type vehicle.

### 3.3. Deduction of Social Costs during the Operation of a Quarry

This study attempted to calculate the social cost or social benefit cost based on the amount of fugitive dust generated by the operation of a quarry. Here, social cost means the costs that society as a whole, including the producer, bears when a certain producer produces a certain good. Based on the social costs of fine and ultrafine dust suggested by the Ministry of Environment (2016), the cost burden of generating PM_10_ is KRW ~285 million per ton. The social cost proposed by the Ministry of Environment (2016) was multiplied by the total annual amount of fugitive dust generated during the operation of a quarry to derive the social cost of a quarry during the railroad life cycle. The quarries operated for 10 h per day; thus, the social cost was estimated to be KRW 36.59 billion.

When calculating the social cost through the amount of fine dust generated by the maintenance vehicles, in the case of the regulator, which generated the largest fugitive emissions, a very high cost was incurred, with an estimation that KRW ~18.73 billion per year of social costs were generated.

## 4. Conclusions and Discussion

In this study, a UAV-based method was used to measure the amount of fine dust generated during the operation of a quarry. The amount of PM generated in the process of collecting and processing raw materials for gravel tracks, which are currently used in many railway lines, was calculated. Based on this figure, the generation of PM before the railway begins operation in the railway life cycle was estimated. In the case of a quarry mine, it is operating 8 h a day, and compared to the annual operating hours of the multiple tie tamper, which operates for the longest among the three types of equipment, it is around 13 times higher than the annual operating hours of a quarry mine. However, fugitive dust emissions were 12 times higher than when a maintenance vehicle was operated on the gravel tracks. Therefore, the quarry mine is using water to reduce fugitive dust. In this study, when fugitive dust occurs, it was confirmed that water is effective in reducing fugitive dust generated in a quarry mine. In future studies, it will be possible to conduct research to reduce the amount of fugitive dust emitted by applying various variables targeting the same quarry mine.

This study determined the amount of PM generated by the collection and processing of raw materials at a quarry during the life cycle of a railway. The amount of PM generated was measured in real time at 1 s intervals using a UAV-based method. A Sniffer4D module, which is a light scattering method for dust measurement with a high sensor reliability, was installed on the UAV. The dust concentrations at the quarry and nearby areas that were difficult for humans to access were measured to determine the total amount of PM generated per year, and then to estimate the social cost.

A move toward sustainable railroad construction and the improvement of railroad facilities has been proposed in Korea, with gravel railroads widely used because of their relatively low noise generation and low initial construction cost. Various environmental problems are caused by the operation of the quarry, which is the first stage of the railroad life cycle. Based on the results of this study, it is necessary to review alternatives that could potentially reduce the amount of dust generated during the operation of a quarry, which is the stage of the railroad life cycle in which the most fugitive dust is generated. An expansion in the use of concrete tracks is a possible response. In addition, it is possible to derive the amount of fugitive dust generated in consideration of the entire life cycle of the railroad through a future study that additionally reviews the matters that take into account the overall flow of the railroad life cycle, such as quarry mine development, crushing, and transportation.

## Figures and Tables

**Figure 1 sensors-21-03206-f001:**
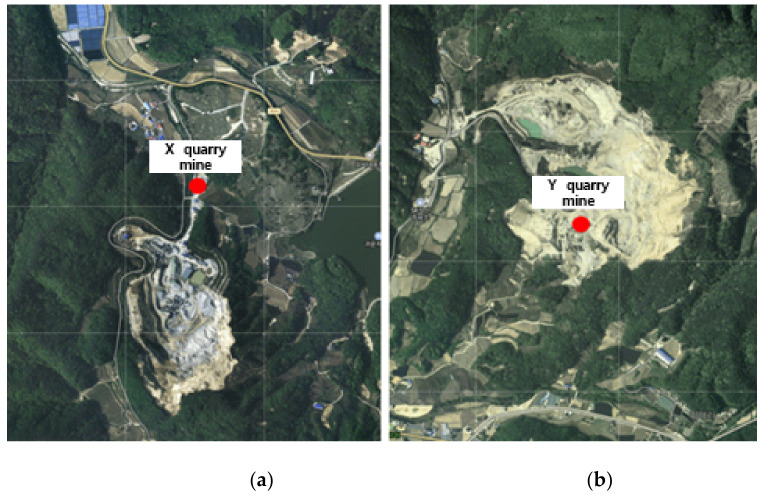
Location of the study sites. (**a**) X quarry; (**b**) Y quarry.

**Figure 2 sensors-21-03206-f002:**
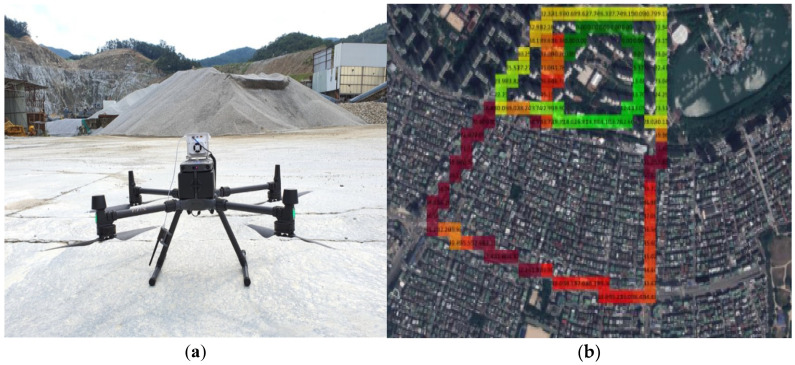
Measurement of the fine particulate matter (PM) concentration using a Sniffer4D module. (**a**) UAV with sniffer 4D; (**b**) 2D maps for data analyzed in Sniffer4D mapper.

**Figure 3 sensors-21-03206-f003:**
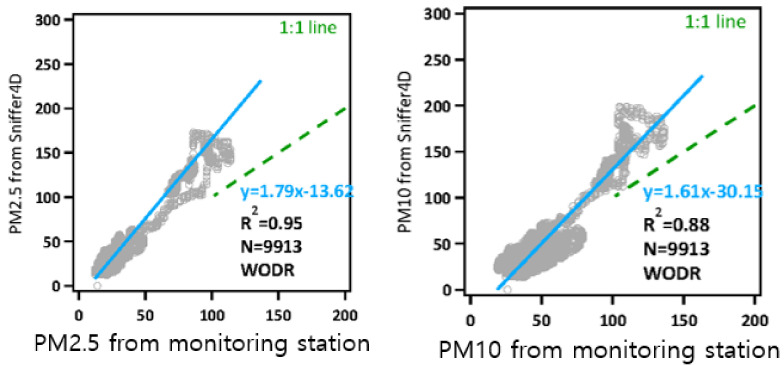
Reliability of the Sniffer4D sensor (PM_2.5_ and PM_10_).

**Figure 4 sensors-21-03206-f004:**
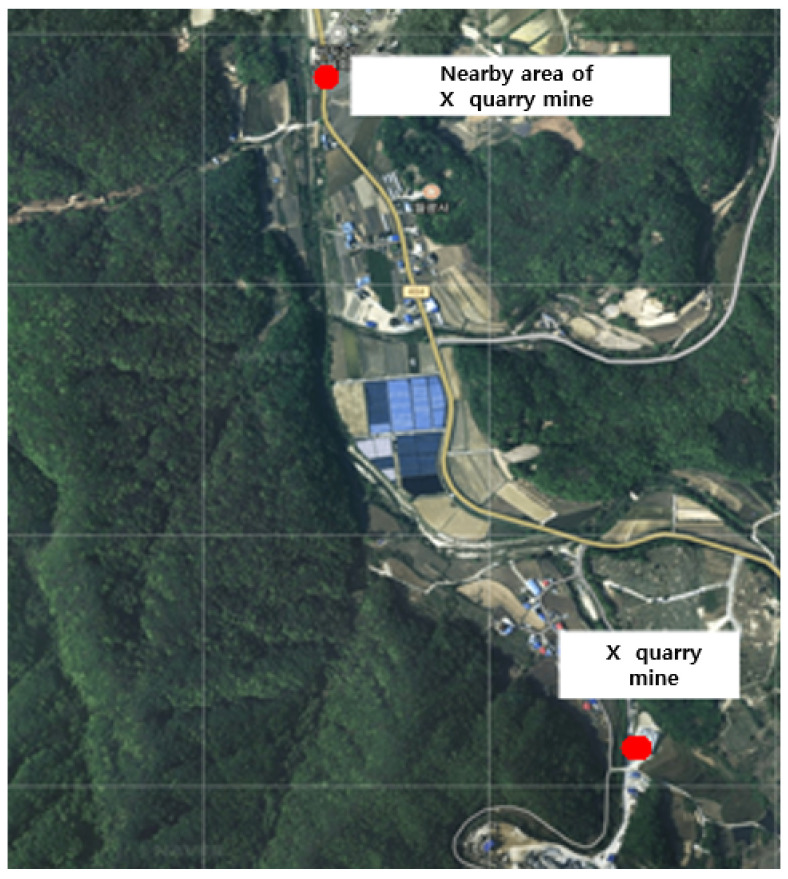
The locations of X quarry and the nearby area used for determining the background concentration.

**Figure 5 sensors-21-03206-f005:**
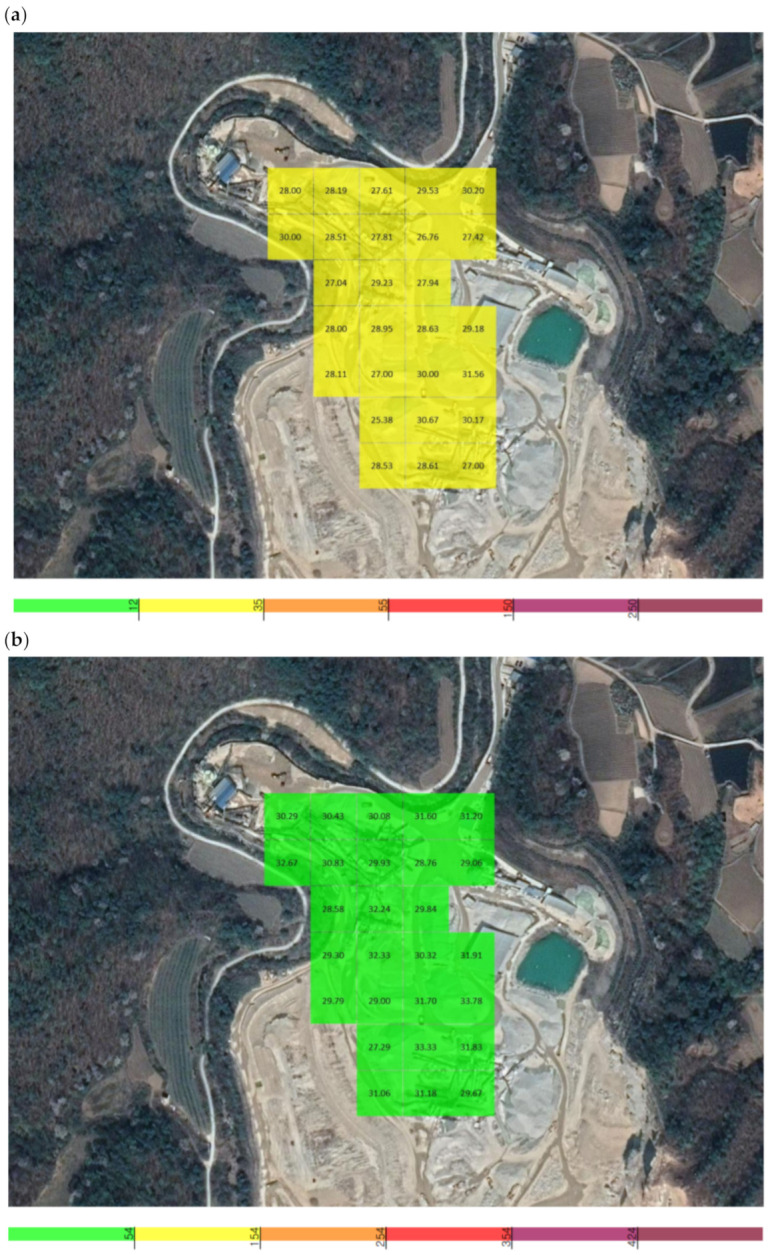
Grid detection area of the unmanned aerial vehicle (UAV) measuring sensor at X quarry: (**a**) PM_2.5_, (**b**) PM_10_.

**Figure 6 sensors-21-03206-f006:**
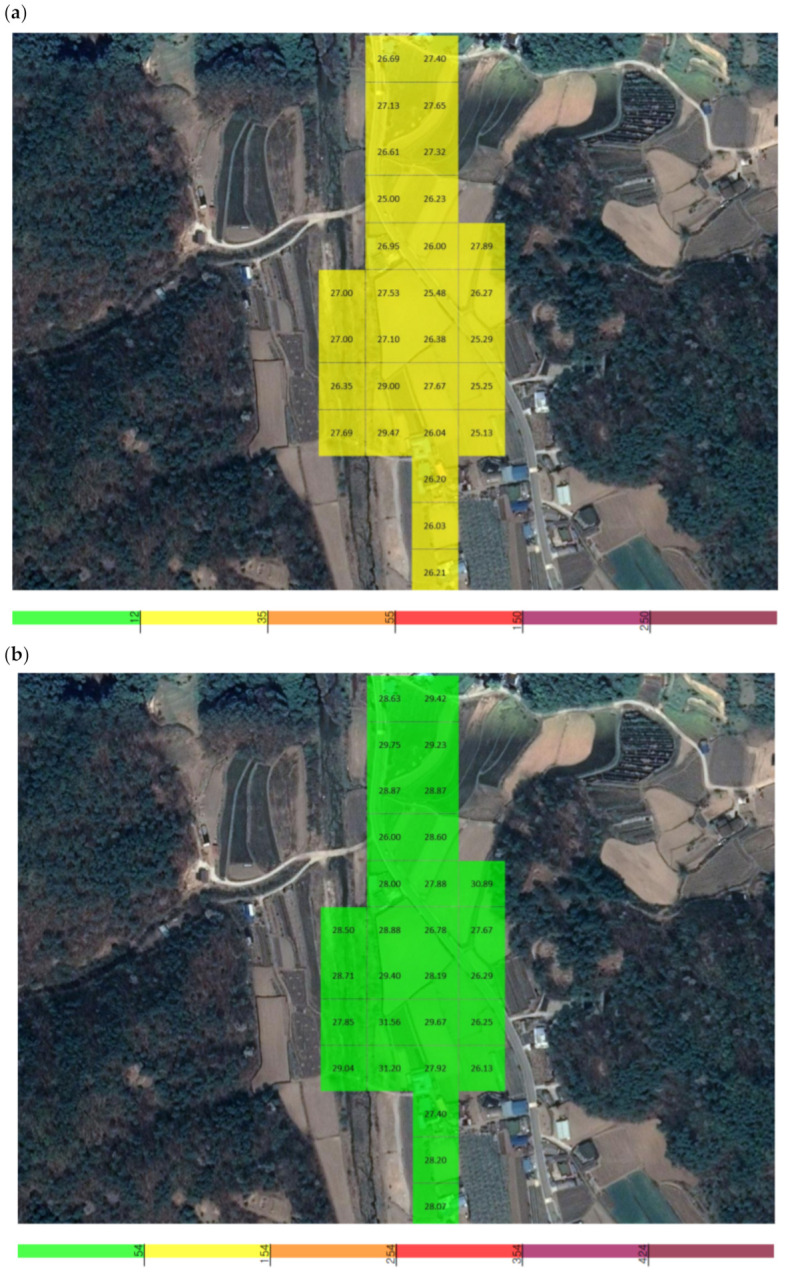
Grid detection area of the UAV measuring sensor in the area nearby X quarry: (**a**) PM_2.5_, (**b**) PM_10_.

**Figure 7 sensors-21-03206-f007:**
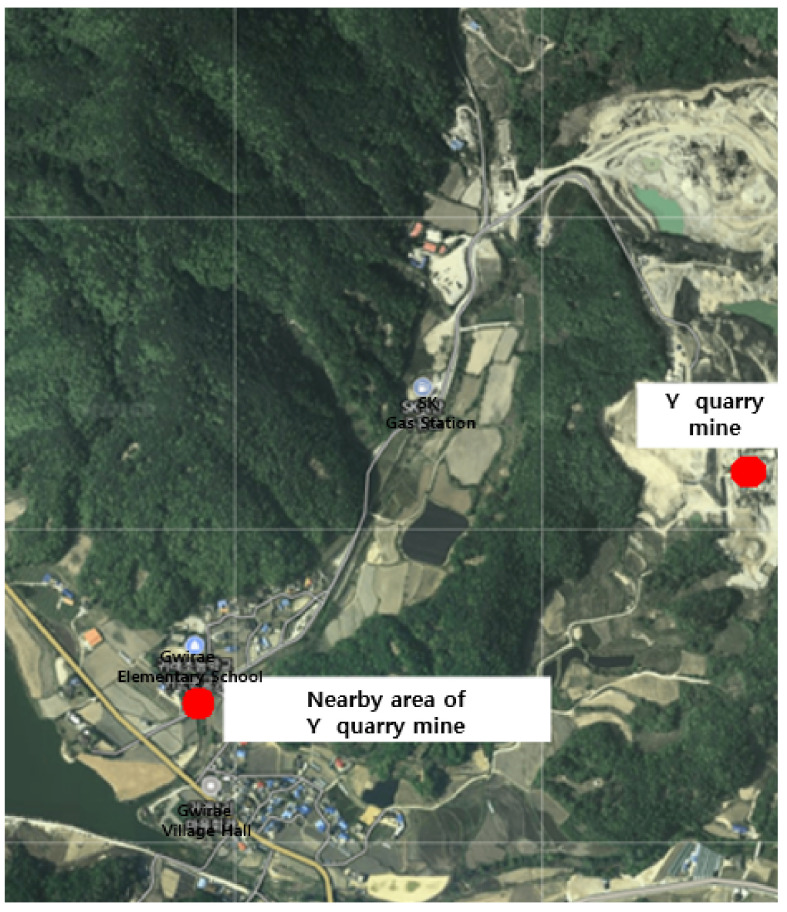
The location of Y quarry and the nearby area used for determining the background concentration.

**Figure 8 sensors-21-03206-f008:**
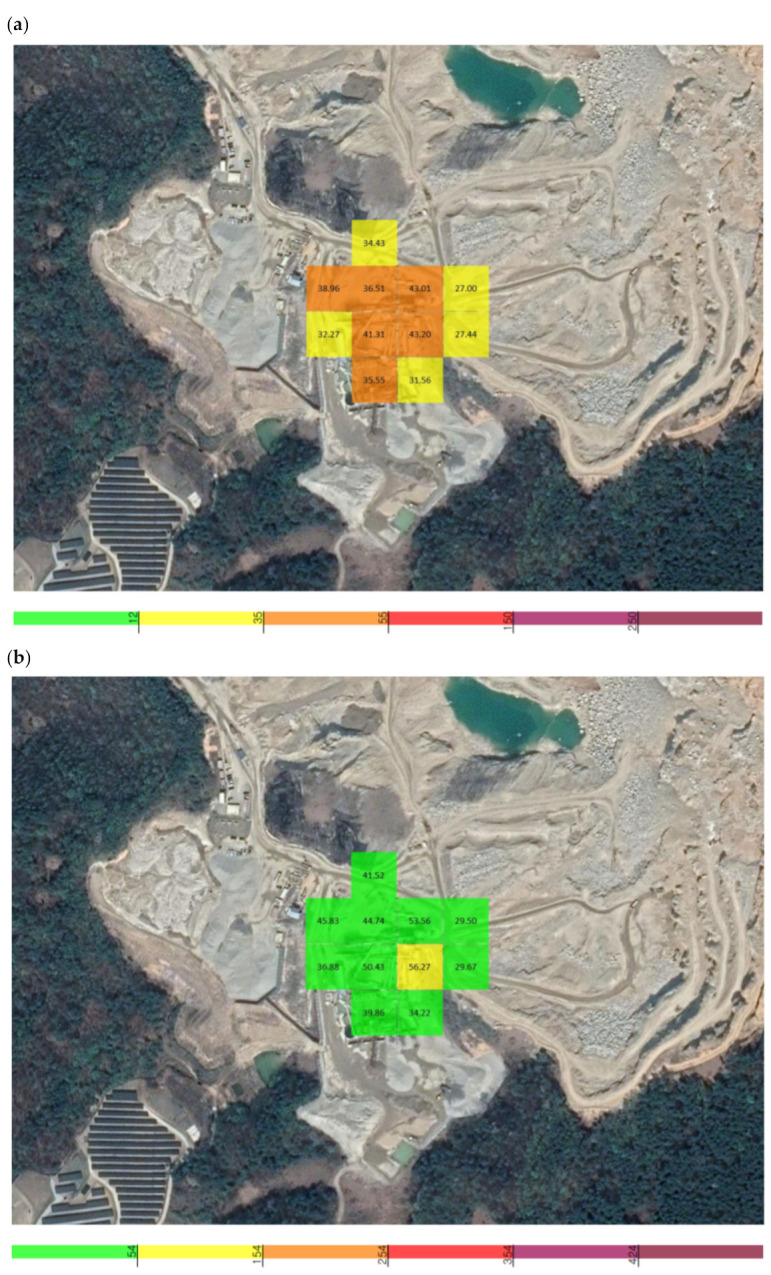
Grid detection area of the unmanned aerial vehicle (UAV) measuring sensor at Y quarry: (**a**) PM_2.5_, (**b**) PM_10_.

**Figure 9 sensors-21-03206-f009:**
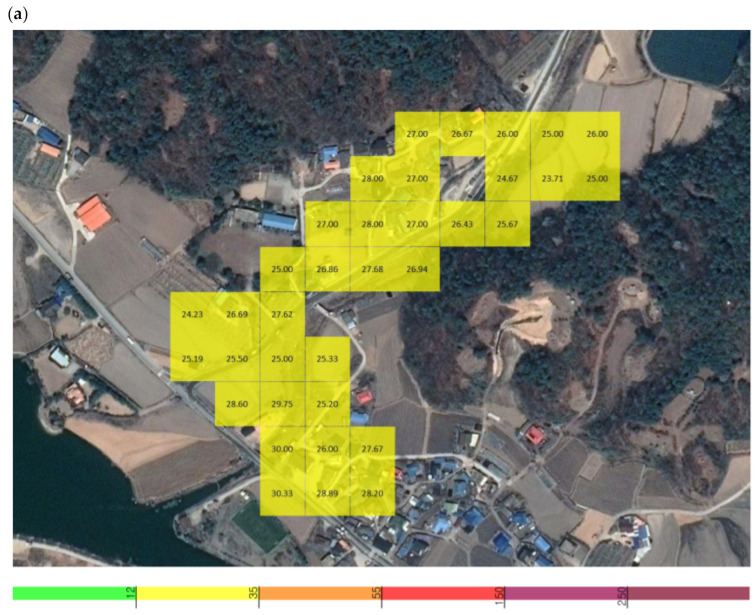
Grid detection area of the UAV measuring sensor in the area nearby Y quarry: (**a**) PM_2.5_, (**b**) PM_10_.

**Table 1 sensors-21-03206-t001:** Calculated fugitive dust generation rates for different dust size classes.

Division	PM2.5	PM10
Measurement area (m^2^)	17,435	
Average wind speed (m/s)	3.6	
Generation rate of fugitive dust (mg/s)	816.0	1192.6
Generation rate of fugitive dust (g/h)	2937.5	4293.2

## Data Availability

Not applicable.

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
