# Peer review of "A UAV-Based Air Quality Evaluation Method for Determining Fugitive Emissions from a Quarry during the Railroad Life Cycle"

_sensors, 2021, doi:10.3390/s21093206_

Round 1

Reviewer 1 Report

In my opinion, the paper is suitable to be published in the Journal as a Communication after some corrections and improvements. The results may be useful for the scientific community and for decision factors. The conclusions should be better explained, what should we do with these results?

 Please find attached my general and specific comments. 

Reviewer 2 Report

Major comment:

L128: I cannot fully follow this discussion. Because the Sniffer4D sensor’s reliability was confirmed for PM2.5 and PM10 as introduced in Section 2.3, why the discussion of “PM1” is abruptly started here? Without the evaluation, the discussion is meaningless. I would like to request to include the evaluation for PM1, or remove all relevant discussion for PM1.

Minor comments:

L 33: Because it is not stated the absolute values of a standard, it is required to put absolute values here for readers.

L71-80: Please re-organize the paragraph. Each paragraph seems to be short.

L131 and related all parts: It is not understandable for this kind of time identification (e.g., 14:6:22 here). Please coordinate as “HH:MM:SS”.

L143: There is no caption for these figures. Moreover, because the resolution is low, we cannot see the exact value from this figure and color-scale. Please re-make this figure.

L196: Again, there is no figure captions here.

Round 2

Reviewer 1 Report

I consider that the manuscript has been improved, I find it useful and suitable for publication. One minor comment: there is still mentioned PM1 in abstract and in text. Please remove that, as it is not relevant in the present form. 

Reviewer 2 Report

I appreciate the authors to revise the manuscript based on my review comment. I have confirmed that all minor points have been addressed. However, there remained one issue.

In my previous review round, I have pointed out the following as “L128: I cannot fully follow this discussion. Because the Sniffer4D sensor’s reliability was confirmed for PM2.5 and PM10 as introduced in Section 2.3, why the discussion of “PM1” is abruptly started here? Without the evaluation, the discussion is meaningless. I would like to request to include the evaluation for PM1, or remove all relevant discussion for PM1.” As major comment.

The authors have replied “As you said, the reliability of the Sniffer 4D sensor was confirmed only by PM2.5 and PM10, so the result corresponding to PM1 was deleted. (line 138‐144, page 6; line 151‐157, page 8; line 182‐188, page 10; line 196‐203; page 12)”.

I think this revision is not enough. The revised manuscript still contained the discussion for PM1 in Section 3.2. If this estimation for fugitive dust is presented, the result should be also presented, or this estimation should be completely removed. Please organize this papers methodology and purpose.

In this revision process, the inserted text seems to use different font or character size. It should be confirmed that the journal style before the submission. 
